# A natural experiment study: Low-profile double plating versus single plating techniques in midshaft clavicle fractures— Study protocol

**Yannic Lecoultre**[1,2]*, **Bryan J. M. van de Wall**[1,2,3], **Nadine Diwersi**[1,3], **Steffen W. Pfarr**[4], **Beat Galliker**[5], **Reto Babst**[1,2], **Björn-Christian Link**[1], **Frank J. P. Beeres**[1]

1 Department of Orthopedic and Trauma Surgery, Lucerne Cantonal Hospital, Lucerne, Switzerland, 2 Faculty of Health Sciences and Medicine, University of Lucerne, Lucerne, Switzerland, 3 Department of Surgery, Obwalden Cantonal Hospital, Sarnen, Switzerland, 4 Department of Surgery, Schwyz Hospital, Schwyz, Switzerland, 5 Department of Surgery, Sursee Hospital, Sursee, Switzerland

* yannic.lecoultre@luks.ch

## Abstract

### Background

Single plate osteosynthesis is commonly employed when performing surgical stabilization of midshaft clavicle fractures. In recent years, a smaller structural low-profile double plating technique has been described as a possible solution for the high removal rates associated with single plating. A previous meta-analysis has demonstrated that low-profile double plating attains the same healing rates as single plating without a higher chance of fracture-related infections. This meta-analysis, however, was based on relatively small studies. Therefore, a multicentre prospective natural experiment was designed using natural variation in treatment regimens and geographical location of the trauma as treatment allocation mechanism to compare both treatments on a larger scale. This manuscript describes its protocol.

### Material & methods

Patients ($\geq$16 years) with primary midshaft clavicle fractures that are eligible for operative treatment will be included. Treatment allocation will be determined by the geographical location of the accident and local hospital providing treatment. In two centres, single plating is the treatment of choice for these patients. In two others, low-profile double plating has become the standard treatment. For the low-profile double plating group, one superiorly positioned VariAx 2.0mm and one anterior VariAx 2.4mm or 2.7mm plate will be used. For the single plating group, the standard locally available implant will be used. A total of 336 patients will be included. The primary outcome of interest is re-intervention. Secondary outcomes include complications, operative time, length of incision, functional scores (DASH, EQ-5D-DL, VAS-Pain/Satisfaction) and cost-effectiveness.

**Data Availability Statement:** No datasets were generated or analysed during the current study. All

relevant data from this study will be made available upon study completion.

**Funding:** This investigator-driven study has been provided with a grant from Stryker to cover the logistical costs (salary study coordinator, data management, costs related to congresses, and publication fees). Initials of the author who received the award: F.B. Grant number: Luzerner Katonsspital T-I-101 Contact details: Stryker Osteonics SA,
Niederlassung Biberist
Burgunderstraße 13
CH-4562 Biberist / SO https://www.stryker.com/ch/de/about/our-locations/biberist.html The funders did not and will not have a role in study design, data collection and analysis, decision to publish, or preparation of the manuscript.

**Competing interests:** The authors have declared that no competing interests exist.

**Abbreviations:** AE, Adverse Event; BMI, Body MASS Index; CI, Confidence Interval; DASH, Disabilities of the arm, shoulder, and hand; EQ, EuroQol; NE, Natural Experiment; OR, Odds Ratio; RCT, Randomized Controlled Trial.

## Discussion

This study will determine whether low-profile double plating has significant clinical and cost-effective benefits over single plating techniques in midshaft clavicle fractures. The study will also give insight in the performance of a natural experiment study design for orthopedic trauma research.

## Trial registration

This study has been registered on ClincialTrials.gov, identifier NCT 05579873.

## Background

Exposed to the exterior environment due to a lack of surrounding muscles or subcutaneous tissue, the clavicle is prone to traumatic injury. Clavicle fractures account for 5 to 6% of all fractures, of which 69 to 81% occur in the midshaft [1, 2]. Surgical stabilization using a single plate, positioned either anteriorly or superiorly, is the most common surgical technique when patients/surgeons decide on operative treatment [3]. The major disadvantage of plate fixation, however, is implant irritation. As a result, up to 64% of patients undergo re-interventional surgery for implant removal [4].

In recent years a low-profile double plating technique with smaller plates has been described as a possible solution to reduce the high rates of implant irritation [5]. Biomechanically, low-profile double plating is expected to be at least as stable as single plating [6]. A recently published meta-analysis showed that low-profile double plating for midshaft clavicle fractures is a safe procedure attaining the same high union rates seen in patients treated with single plating [7]. Additionally, double plating appears to have a lower overall complication and reintervention rate, mainly due to the lower incidence of implant-related complaints. Low-profile dual plating, however, is a relatively new technique and should be further explored to test whether our findings can be confirmed.

Although randomized controlled trials (RCTs) are considered the gold standard for testing the effectiveness of such new interventions, the artificial conditions that are usually imposed on surgical practice frequently limit the feasibility and the generalizability of its results. A natural experiment (NE) design offers a possible solution. NEs are observational studies in which patients are exposed to either the experimental or the control condition, and treatment allocation is determined by factors outside the control of the investigators (e.g., geographical location) [8]. Although not the same, the process governing treatment allocation arguably resembles that of randomization, since treatment allocation is expected to be (to a large extent) independent of patients' characteristics and their prognosis.

This protocol encompasses a prospective NE comparing low-profile double plating to single plating in midshaft clavicle fractures. The primary endpoint is the overall re-intervention rate, with a specific interest in implant removal due to irritation. Secondary endpoints include all complications, healing, the general quality of life, functional results, and cost-effectiveness.

## Methods/design

### Aim

To compare single plating to low-profile double plating in midshaft clavicle fractures with regard to re-intervention, complications, healing, quality of life, functional results, and cost-effectiveness.

## Study design

This study will be a multicenter prospective cohort study between four Swiss centers with comparable patient case-mix. At which hospital/surgeon trauma patients are presented after an accident depends on the geographical location of the accident. We hypothesize that different Swiss centers treat a similar caseload of trauma patients with clavicle fracture and that the patient case-mix is similar across the different centers.

Of the four participating centers, two centers are specialized in single plating, and two are specialised in low-profile double plating will participate. Treatments will be naturally allocated by the geographical location of the accident as shown in Fig 1. Although centers are divided

| | STUDY PERIOD | | | | |
|---|---|---|---|---|---|
| | Allocation | Enrolment | Post-allocation | | Close-out |
| **TIMEPOINT** | Time of accident | Within 2 weeks from accident | *3 months* | *12 months* | *2 years* |
| **ENROLMENT:** | | | | | |
| **Eligibility screen** | | X | | | |
| **Informed consent** | | X | | | |
| **Allocation by accident location** | X | | | | |
| **INTERVENTIONS:** | | | | | |
| **Single plate osteosynthesis** | | X | | | |
| **Double plate osteosynthesis** | | X | | | |
| **ASSESSMENTS:** | | | | | |
| *X-ray* | X | | X | X | |
| *DASH score** | | X | X | X | |
| *EQ5D score** | | X | X | X | |
| *VAS-pain and satisfaction** | | | X | X | |
| *Re-interventions* | | | | X | X |
| *Complications* | | | X | X | X |

**Fig 1. Schedule of enrolment, interventions, and assessments.** *Not part of standard care.

into either category based on their standard treatment of care, surgeons will be allowed to alter their treatment plan based on their own surgical expertise and according to what they (after consultation of the patient) consider the most appropriate treatment. This study design is sometimes referred to as a NE and is arguably less prone to confounding bias than conventional observational studies, provided the above-mentioned assumptions hold.

### Participant selection

**Eligibility criteria.** All adult patients (>18 older) presenting at the emergency department (ED) or outpatient clinic with midshaft clavicle fractures (Robinson Type II or AO/OTA 15.2) [9] will be included. In-and exclusion criteria are shown in Fig 2. Patients will be screened for eligibility by the treating physician at presentation to the emergency department or during outpatient visits (within two weeks after trauma).

**Participant recruitment and screening.** Patients will be asked to participate in the study after the established decision on operative treatment. If eligible, the treating physician or local investigator will explain to each participant the nature of the study, its purpose, the procedures involved, the expected duration, the potential risks and benefits, and any discomfort it may entail. Each participant will be informed that participation in the study is voluntary, that he or

## INCLUSION CRITERIA

- Aged 18 years or older.
- Written informed consent.
- Primary midshaft clavicle fracture (Robinson Type II or AO 15.2).
- Patients that are eligible for operative treatment of clavicle fracture, which may include but is not limited by:
  - Displacement of one or more shaft width.
  - Shortening of more than 1cm in length.
  - High physical activity level.

## EXCLUSION CRITERIA

- Delayed presentation (>14 days).
- Initial operative treatment at a non-participating hospital.
- Open fractures.
- Pathological fractures.
- Re-fractures of the clavicle.
- Cognitive impairment or language barrier precluding answering questionnaires.
- Unable to complete follow-up (e.g. different residential area/tourist).

**Fig 2. In/exclusion criteria.**

she may withdraw from the study at any time, and that withdrawal of consent will not affect his or her subsequent medical assistance and treatment. The participant will be informed that their pseudonymized medical records may be examined by authorized researchers other than their treating physician.

**Data management.** Data will be stored in pseudonymized form in an encrypted online database by the local investigators, which can access their respective individual patient data. Full access to the database is granted to the study coordinator.

## Study interventions

**Low-profile double plating group.** Low-profile double plating consists of one Stryker VariAx 2.0mm plate positioned on the superior aspect of the clavicle and a second Stryker VariAx 2.4mm or 2.7mm on the anterior side. Use of this implant will be according to the manufacturer's indication of use.

**Single plating group.** The choice of implant used for single plating is left at the discretion of the treating surgeon and the availability of implants in participating centers. Possible implants consist of 2.7 and 3.5mm plates placed on the superior or anterior side.

**Standardization of care.** Preoperative care will consist of general anesthesia and routine antibiotic prophylaxis. Incision and cutaneous nerve sparing are left at the treating surgeon's discretion and will be mentioned in the operation report. In case of a multi-fragmentary fracture pattern, bridge plating will be performed. In case of a simple fracture, a neutralization plate (with or without lag screws) or a compression plate will be used. Closure of deltopectoral fascia with absorbable braided sutures and skin with nonabsorbable monofilament sutures (Allgöwer-technique) will be performed for all patients. Operations will either be performed or directly supervised by a specialized trauma surgeon. Aftercare comprises a sling for comfort for 1–2 weeks, no weight bearing for six weeks, and outpatient clinic visits, including AP and axial clavicle X-rays at six weeks, three months, and one year after surgery in both treatment groups.

**Study outcomes.** The primary outcome of this study is the number of re-interventions after two years of follow-up. Secondary outcomes include the number of re-interventions at one year of follow-up, all complications such as fracture-related infection [10], (a)symptomatic non-union, numbness below the scar, and self-reported implant irritations, disability of the shoulder and arm (DASH) score, EuroQol (EQ)-5D score, and the visual analogue scale (VAS) score for pain and patient satisfaction at 3, 12, and 24 months follow-up. A complete overview of the primary and secondary outcomes can be found in Table 1.

**The outcome measure, follow-up and timeline.** At baseline, the treating physician or local investigator will collect the following characteristics: age, gender, body mass index (BMI), smoking status, comorbidities, medication, trauma mechanism, concomitant injuries and planned operative procedure (including treating hospital and surgeon), pre-injury DASH and EQ-5D-5L. The x-ray at presentation or outpatient clinic visit will be used to determine fracture type and dislocation. Fracture type will be determined according to the AO/OTA classification system for midshaft clavicle fractures [9]. Patients will receive the operation within two weeks of presentation. All operative data including interval between trauma and surgery, level of experience of surgeon, type of plate used, plate position, number and type of screws, type and length of incision and surgery duration will also be collected after the surgical procedure.

The patients will return to the outpatient clinic at three and twelve months. The outpatient clinic visit will include several questionnaires (e.g., VAS, EQ5D, DASH), x-rays, and a physical examination (Fig 1). All data and adverse events (AE) will be registered. At two years, the patient will be either seen at the outpatient clinic visit or interviewed by telephone, and the

**Table 1. Primary and secondary outcomes.**

| Primary Objective |
| --- |
| Re-intervention (all indications) after 2-year follow-up |

| Secondary Objective |
| --- |
| Re-intervention rate (including implant removal) after 1-year follow-up. |
| All other complications (including complications treated non-operatively) <br> • Fracture related infection. <br> • Symptomatic non-union defined as absence of radiological signs of healing (callus formation or fading of fractures lines) combined with pain at the fracture site at 9 months. <br> • Asymptomatic non-union defined as absence of radiological signs of healing (callus formation or fading of fracture lines) without any clinical symptoms. <br> • Numbness below scar line related postoperatively and at 12-months follow-up. <br> • Self-reported implant irritation or other reasons for implant removal at 12 months. |
| Operative time. |
| Length of surgical incision. |
| QuickDASH score at baseline (pre-injury), 3- and 12-months follow-up. |
| EQ-5D-5L at baseline (pre-injury), 3- and 12-months follow-up. |
| VAS pain and patient satisfaction at 3- and 12-months follow-up. |

electronic patient database will be accessed to assess patient-reported outcome measures (PROMS), complications and reintervention rate.

Healing is defined as the presence of bridging bone on 3 of 4 cortices or, in cases of direct fracture healing, disappearance of the fracture line on ap and tangential x-ray views. It is assessed on follow-up x-rays 3- and 12-months after surgery. All subsequent surgical procedures during follow-up at the same site after the index operation (including hardware removal) are defined as re-interventions [11]. Complications are defined and classified according to the Clavien-Dindo-Classification of Surgical Complications [12].

All the measurements and examinations described are part of standard care, except the questionnaires. The questionnaires will be filled in at regular outpatient clinic visits or by telephone interview.

## Sample size calculation

The sample size calculation is based on the primary outcome (re-intervention). Based on a recent meta-analysis of our study group, a risk reduction of 10% is expected in favor of low-profile double plating compared to single plating. To demonstrate this difference using a chi-square-test, (two-sided alpha = 0.05, power = 0.8) approximately 152 patients per treatment group are needed. To compensate for a potential 5% loss to follow-up, a total of 336 patients are required.

## Statistical analysis

The statistical package IBM SPSS Statistics for Windows, version 28.0, will be used for analysis. Depending on the normality distribution, baseline characteristics will be described as means and standard deviations or median and interquartile range for continuous variables. Categorical variables will be reported in counts and percentages. Differences between treatment groups will be analyzed with an independent samples student's T-test or Mann-Whitney-U test (depending on distribution). Categorical variables will be analyzed using the Chi-square test.

The primary outcome will be analyzed using logistic regression with revision surgery as the dependent and treatment as the independent variable. The relative risk (RR) will be calculated with a 95% confidence interval (95% CI). Multivariable logistics regression analysis will also be performed to account for potential known confounders including age and fracture type.

Repeatedly measured outcomes (DASH and EQ5D) will be analyzed using a regression analysis with the scores as dependent and treatment as independent variable. The potential confounders, age and fracture type, will also be included as covariates in the model. Regression coefficients will be calculated with corresponding 95% CI. Missing data (caused by incomplete hospital documentation or loss-to-follow-up) will be addressed using multiple imputation techniques.

## Ethics approval

This study was ethically approved by the Northwestern and central Swiss ethics commission. The identifier number is 2022–00574

## Discussion

Previously, a meta-analysis on single vs low-profile double plating demonstrated that low-profile double plating attains the same healing rates as single plating without a higher chance of fracture-related infections [7]. This study however, was based on studies with relatively small sample sizes. The present NE study will provide the required evidence to confirm or revoke the aforementioned conclusions based on a more extensive study population.

Additionally, this study will form the basis for a more prominent use of the NE design in the surgical research. Due to its design, this study has a high feasibility with minimal burden to the participants, since the received care is that of daily clinical practice. Not only will the study help us understand if low-profile double plating has significant clinical and cost-effective benefits over single plating techniques in midshaft clavicle fractures, but it also provides new insights into how the NE design functions in a research field where it has rarely been used before.

## Supporting information

**S1 File. SPIRIT checklist.**
(PDF)

**S2 File. STROBE checklist.**
(PDF)

**S3 File. PLOS ONE clinical studies checklist.**
(PDF)

**S4 File. Inclusion form.**
(PDF)

**S5 File. Inclusion form (English translation).**
(PDF)

**S6 File.**
(PDF)

## Author Contributions

**Conceptualization:** Bryan J. M. van de Wall, Frank J. P. Beeres.

**Funding acquisition:** Bryan J. M. van de Wall, Frank J. P. Beeres.

**Investigation:** Yannic Lecoultre, Steffen W. Pfarr, Beat Galliker.

**Methodology:** Yannic Lecoultre, Bryan J. M. van de Wall, Frank J. P. Beeres.

**Project administration:** Yannic Lecoultre.

**Writing – original draft:** Yannic Lecoultre, Bryan J. M. van de Wall, Nadine Diwersi, Frank J. P. Beeres.

**Writing – review & editing:** Reto Babst, Björn-Christian Link, Frank J. P. Beeres.

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
