## [Decision Letter · Decision Letter 0]

13 Jul 2023

PONE-D-23-13255A natural experiment study: Low-profile double plating versus single plating techniques in midshaft clavicle fractures - Study protocolPLOS ONE

  Dear Dr. Lecoultre,

Thank you for submitting your manuscript to PLOS ONE. After careful consideration, we feel that it has merit but does not fully meet PLOS ONE’s publication criteria as it currently stands. Therefore, we invite you to submit a revised version of the manuscript that addresses the points raised during the review process.Please, take into consideration the proposals of the statistical reviewer (reviewer Nr 3) and address her advices.It is a problem to give a recommendation (accept, reject or revision) on a study protocol, which, obviously does not present any single result. It may be one drop more in the ocean of publications. We hope to receive a manuscript after completion of the study. 

We look forward to receiving your revised manuscript.

Kind regards,

Hans-Peter Simmen, M.D., Professor of Surgery

Academic Editor

PLOS ONE

3. We note that the original protocol that you have uploaded as a Supporting Information file contains an institutional logo. As this logo is likely copyrighted, we ask that you please remove it from this file and upload an updated version upon resubmission.

Reviewers' comments:

Reviewer's Responses to Questions

**Comments to the Author**

1. Does the manuscript provide a valid rationale for the proposed study, with clearly identified and justified research questions?

Reviewer #1: Yes

Reviewer #2: Yes

Reviewer #3: Yes

2. Is the protocol technically sound and planned in a manner that will lead to a meaningful outcome and allow testing the stated hypotheses?

Reviewer #1: Partly

Reviewer #2: Yes

Reviewer #3: Yes

3. Is the methodology feasible and described in sufficient detail to allow the work to be replicable?

Reviewer #1: Yes

Reviewer #2: Yes

Reviewer #3: Yes

4. Have the authors described where all data underlying the findings will be made available when the study is complete?

Reviewer #1: No

Reviewer #2: No

Reviewer #3: No

5. Is the manuscript presented in an intelligible fashion and written in standard English?

Reviewer #1: Yes

Reviewer #2: Yes

Reviewer #3: Yes

6. Review Comments to the Author

You may also provide optional suggestions and comments to authors that they might find helpful in planning their study.

Reviewer #1: please give the clear definition of the terms:: re-interventions, healing and complications?

How exactly are these terms defined at what point in time? What influence do they have on fracture healing and why? How long is the follow-up?

In addition, inclusion and exclusion criteria are not known. What about open fractures? Polytrauma patients? pre-existing conditions? infections? Osteoporosis? Infection as a cause of non-healing?

Duration of surgery? Status of the operator? Number of screws used? Time of accident to operation? Postoperative procedure? Will this be the same for all? immobilization? Suture technique? (Intracutaneous vs. donati?)

Fracture classification?

there also needs to be a clear definition of when the fracture is considered healed and how this is determined and when.

Reviewer #2: 1) General comment

The authors would like to show in their study protocoll “Low-profile double plating versus single plating techniques in midshaft clavicule fractures” clinical and cost-effective benefits between double plating and single plating techniques. They also are trying to gibe insights in the performance of a natural experiment study design for orthopedic trauma research..

The study design is very well thought out or worked out for the questions mentioned. However, it is worth discussing whether the phase of drafting a study design is the right time to publish. At the current time, an infinite number of works that have been carried out are being published. The entire field of medicine is flooded with publications, so that there is not enough time to evaluate the relevant publications. If every protocol of future studies is published now, the flood of publications will only increase. Ultimately, the protocols have not yet proven any essential facts regarding innovations, new therapeutic approaches or new findings.

In any case, it is interesting whether or not the double plating technique has any really significant clinical and cost-effective advantages over the single plating application. This study can certainly be carried out in the setting mentioned and will show corresponding well-founded results. However, it is necessary to wait until the results of the content are available before one wants to expect this from a broader readership. However, if one publishes future study protocols, this will lead to an infinitely progressive increase in publications

If you look at the most famous collection of scientific reports, namely PubMed, around 500,000 documents are published there every year. This entire medical collection is growing so rapidly that one should focus on essential results. This database now includes more than 32 million "citations" from scientific journal articles and online literature if you only look at the biomedical literature (PubMed Help).

2) Specific suggestions

The study mentioned above is well designed and will be appropriate for the intended research question and outcome results

Reviewer #3: In this study protocol, a non-randomized multicenter prospective clinical trial is being proposed to determine if low-profile double plating shows significant clinical and cost-effective benefits in comparison to single plating techniques. The primary outcome is re-intervention rate. Secondary outcomes are complications, operative time, length of incision, function scores, and cost-effectiveness. The target sample size is 336.

Minor revisions:

1- Table at line 158: inclusion is misspelled.

2- Line 167: Indicate if the alpha level was one or two-sided. A single sided test seems appropriate since the low-profile double plating is expected to outperform the single plating technique.

3- For outcomes that are repeatedly measured, statistical methods appropriate for analyzing repeated measures should be used.

7. PLOS authors have the option to publish the peer review history of their article (what does this mean?). If published, this will include your full peer review and any attached files.

Reviewer #1: No

Reviewer #2: No

Reviewer #3: No

---

## [Author Response · Author response to Decision Letter 0]

16 Aug 2023

Journal requirements

- We have re-checked PLOS One's style requirements and adapted our titlepage, formatting, figure captions and file naming.

- We have moved the ethics statement to the methods section.

3. We note that the original protocol that you have uploaded as a Supporting Information file contains an institutional logo. As this logo is likely copyrighted, we ask that you please remove it from this file and upload an updated version upon resubmission.

- Thank you for the notice, we removed all institutional logos from the supporting information files.

- We have re-checked our reference list and replaced reference nr. 8, since it is not listed in medline. Also, we added reference 11 and 12 to meet reviewer 2's requirements.

Comments to the Author

Reviewer #1: please give the clear definition of the terms: re-interventions, healing and complications?

How exactly are these terms defined at what point in time? What influence do they have on fracture healing and why? How long is the follow-up?

In addition, inclusion and exclusion criteria are not known. What about open fractures? Polytrauma patients? pre-existing conditions? infections? Osteoporosis? Infection as a cause of non-healing?

Duration of surgery? Status of the operator? Number of screws used? Time of accident to operation? Postoperative procedure? Will this be the same for all? immobilization? Suture technique? (Intracutaneous vs. donati?)

Fracture classification?

there also needs to be a clear definition of when the fracture is considered healed and how this is determined and when.

- Thank you for your detailed and valuable feedback. We fully agree with your comments. The inclusion criteria, outcome definitions, timepoint of outcome measurement and follow-up duration are described in figures 1, 2 and table 1. We also added a section on operative data that will be collected after the surgical procedure and a description of fracture classification used. Please let us know if you have any additional wishes.

Reviewer #2: 1) General comment

The authors would like to show in their study protocoll “Low-profile double plating versus single plating techniques in midshaft clavicule fractures” clinical and cost-effective benefits between double plating and single plating techniques. They also are trying to gibe insights in the performance of a natural experiment study design for orthopedic trauma research..

The study design is very well thought out or worked out for the questions mentioned. However, it is worth discussing whether the phase of drafting a study design is the right time to publish. At the current time, an infinite number of works that have been carried out are being published. The entire field of medicine is flooded with publications, so that there is not enough time to evaluate the relevant publications. If every protocol of future studies is published now, the flood of publications will only increase. Ultimately, the protocols have not yet proven any essential facts regarding innovations, new therapeutic approaches or new findings.

In any case, it is interesting whether or not the double plating technique has any really significant clinical and cost-effective advantages over the single plating application. This study can certainly be carried out in the setting mentioned and will show corresponding well-founded results. However, it is necessary to wait until the results of the content are available before one wants to expect this from a broader readership. However, if one publishes future study protocols, this will lead to an infinitely progressive increase in publications

If you look at the most famous collection of scientific reports, namely PubMed, around 500,000 documents are published there every year. This entire medical collection is growing so rapidly that one should focus on essential results. This database now includes more than 32 million "citations" from scientific journal articles and online literature if you only look at the biomedical literature (PubMed Help).

2) Specific suggestions

The study mentioned above is well designed and will be appropriate for the intended research question and outcome results

- First, we want to thank you for your appreciative feedback on our study approach. We agree with you that the public libraries are being flooded with publications. The quality of these studies also varies considerably. This is however precisely the reason why we think it is important to publish protocols prior to conducting the study itself including those from prospective observational studies. This creates transparency on the study design making it easier for readers to assess the quality and validity of the end results; a feature many publications are lacking. We agree that the number of publications should be reduced in such a way that only good quality data finds its way to our public libraries. Protocols, in our opinion, are the gatekeepers for quality and should not be seen as just an additional publication but rather a sign or mark of quality when the end results do appear. We hope you may understand our point of view and support the publication of this protocol. 

Reviewer #3: In this study protocol, a non-randomized multicenter prospective clinical trial is being proposed to determine if low-profile double plating shows significant clinical and cost-effective benefits in comparison to single plating techniques. The primary outcome is re-intervention rate. Secondary outcomes are complications, operative time, length of incision, function scores, and cost-effectiveness. The target sample size is 336.

Minor revisions:

1- Table at line 158: inclusion is misspelled.

- Thank you for pointing this out. We corrected this mistake.

2- Line 167: Indicate if the alpha level was one or two-sided. A single sided test seems appropriate since the low-profile double plating is expected to outperform the single plating technique.

- This was a two sided alpha level. Indeed, based on prior studies we expect that the reintervention rate will be lower among the double plating group. A one-sided test would theoretically be correct. However, we are not absolutely certain this will be the case. For this reason we prefer to play it safe and taking this uncertainty into account by using a two-sided alpha in our sample size calculation. 

3- For outcomes that are repeatedly measured, statistical methods appropriate for analyzing repeated measures should be used.

- Thank you for the input, we now specified this in the statistical analysis section.

---

## [Editor Report · Decision Letter 1]

25 Aug 2023

A natural experiment study: Low-profile double plating versus single plating techniques in midshaft clavicle fractures - Study protocol

PONE-D-23-13255R1

Dear Dr. Lecoultre,

We’re pleased to inform you that your manuscript has been judged scientifically suitable for publication and will be formally accepted for publication once it meets all outstanding technical requirements.

Kind regards,

Hans-Peter Simmen, M.D., Professor of Surgery

Academic Editor

PLOS ONE
---

## [Editor Report · Acceptance letter]

31 Aug 2023

PONE-D-23-13255R1 

A natural experiment study: Low-profile double plating versus single plating techniques in midshaft clavicle fractures - Study protocol 

Dear Dr. Lecoultre:

I'm pleased to inform you that your manuscript has been deemed suitable for publication in PLOS ONE. Congratulations! Your manuscript is now with our production department. 

Kind regards, 

on behalf of

Dr. Hans-Peter Simmen 

Academic Editor

PLOS ONE